# The Twitter Social Mobility Index:
# Measuring Social Distancing Practices from Geolocated Tweets

**Paiheng Xu, Mark Dredze**
Malone Center for Engineering in Healthcare
Center for Language and Speech Processing
Department of Computer Science
Johns Hopkins University
`paiheng,mdredze@jhu.edu`

**David A. Broniatowski**
Department of Engineering Management
and Systems Engineering
Institute for Data, Democracy, and Politics
The George Washington University
`broniatowski@gwu.edu`

## Abstract

Social distancing is an important component of the response to the novel Coronavirus (COVID-19) pandemic. Minimizing social interactions and travel reduces the rate at which the infection spreads, and "flattens the curve" such that the medical system can better treat infected individuals. However, it remains unclear how the public will respond to these policies. This paper presents the Twitter Social Mobility Index, a measure of social distancing and travel derived from Twitter data. We use public geolocated Twitter data to measure how much a user travels in a given week. We find a large reduction in travel in the United States after the implementation of social distancing policies, with larger reductions in states that were early adopters and smaller changes in states without policies. Our findings are presented on `http://socialmobility.covid19dataresources.org` and we will continue to update our analysis during the pandemic.

## 1 Introduction

The outbreak of the SARS-CoV-2 virus, a Coronavirus that causes the disease COVID-19, has caused a pandemic on a scale unseen in a generation. Without an available vaccine to reduce transmission of the virus, public health and elected officials have called on the public to practice social distancing. Social distancing is a set of practices in which individuals maintain a physical distance so as to reduce the number of physical contacts they encounter (Maharaj and Kleczkowski, 2012; Kelso et al., 2009). These practices include maintaining a distance of at least six feet and avoiding large gatherings (Glass et al., 2006). At the time of this writing, in the United States nearly every state has implemented state-wide "stay-at-home" orders to enforce social distancing practices (Zeleny, 2020).

While an important tool in the fight against COVID-19, the implementation of social distancing by the general public can vary widely. While a state governor may issue an order for the practice, individuals in different states may respond in different ways. Understanding actual reductions in travel and social contacts is critical to measuring the effectiveness of the policy. These policies may remain in effect for an extended period of time. Thus, the public may begin to relax their practices, making additional policies necessary. Additionally, epidemiologists already model the impact of social distancing policies on the course of an outbreak (Prem et al., 2020; Fenichel et al., 2011; Caley et al., 2008). These models may be more effective when incorporating actual measures of social distancing, rather than assuming official policies are implemented in practice.

It can be challenging to obtain data on the efficacy of social distancing practices, especially during an ongoing pandemic. A recent Gallup poll surveyed Americans to find that many adults are taking precautions to keep their distance from others (Saad, 2020). However, while polling can provide insights, it cannot provide a solution. Polling is relatively expensive, making it a poor choice for ongoing population surveillance practices and providing data on specific geographic locales, i.e. US States and major cities (Dredze et al., 2016a). Additionally, polling around public health issues suffers from response bias, as individuals may overstate their compliance with established public health recommendations (Adams et al., 1999).

Over the past decade, analyses of social media and web data have been widely adopted to support public health objectives (Paul and Dredze, 2017). In this vein, several efforts have emerged over the past few weeks to track social distancing practices using these data sources. Google has released "COVID-19 Community Mobility Reports"

which use Google data to "chart movement trends over time by geography, across different categories of places such as retail and recreation, groceries and pharmacies, parks, transit stations, workplaces, and residential" (Google, 2020). The Unacast "Social Distancing Scoreboard" uses data collected from 127 million monthly active users to measure the implementation of social distancing practices (Unacast, 2020). Researchers at the Institute for Disease Modeling have used data from Facebook's "Data for Good" program to model the decline in mobility in the Greater Seattle area and its effect on the spread of COVID-19 (Burstein et al., 2020). Using cell phone data, the New York Times completed an analysis that showed that stay-at-home orders dramatically reduced travel, but that states that have waited to enact such orders have continued to travel widely (Glanz et al., 2020). These efforts provide new and important opportunities to study social distancing in real-time.

We present the Twitter Social Mobility Index, a measure of social distancing and travel patterns derived from public Twitter data. We use public geolocated Twitter data to measure how much a user travels in a given week. We compute a metric based on the standard deviation of a user's geolocated tweets each week, and aggregate these data over an entire population to produce a metric for the United States as a whole, for individual states and for some US cities. We find that, taking the US as a whole, there has been a dramatic drop in travel in recent weeks, with travel between March 16 and April 27, 2020 showing the lowest amount of travel since January 1, 2019, the start of our dataset. Additionally, we find that travel reductions are not uniform across the United States, but vary from state to state. However, there's no clear correlation between the social mobility and confirmed COVID-19 cases at the state level. A key advantage of our approach is that, unlike other travel and social distancing analyses referenced above, we rely on entirely public data, enabling others to replicate our findings and explore different aspects of these data. Additionally, since Twitter contains user generated content in addition to location information, future analyses can correlate attitudes, beliefs, and behaviors with changes in social mobility.

Our findings are presented on `http://socialmobility.covid19dataresources.org` and we will continue to update our analysis during the pandemic.

## 2 Data

Twitter offers several ways in which a user can indicate their location. If a user is tweeting from a GPS enabled device, they can attach their exact coordinate to that tweet. Twitter may then display to the user, and provide in their API, the specific place that corresponds to these coordinates. Alternatively, a user can explicitly select a location, which can be a point of interest (coffee shop), a neighborhood, a city, state, or country. If the tweet is public, this geolocation information is supplied with the tweet.

We used the Twitter streaming API[1] to download tweets based on location. We used a bounding box that covered the entire United States, including territories. We used data from this collection starting on January 1, 2019 and ending on April 27, 2020. In total, this included 3,768,959 Twitter users and 469,669,925 tweets in United States.

## 3 Location Data

We process the two types of geolocation information described in the previous section.

**Coordinates** The exact coordinates (latitude/longitude) provided by the user ("coordinates" field in the Twitter JSON object). About 8% of our data included "coordinates".

**Place** The "place" field in the Twitter json object indicates a known location in which the tweet was authored. A place can be a point of interest (a specific hotel), a neighborhood ("Downtown Jacksonville"), a city ("Kokomo, IN"), a state ("Arizona") or a country ("United States"). The place object contains a unique ID, a bounding box, the country and a name. More information about the location is available from the Twitter GEO API. A place is available with a tweet in either of two conditions. First, Twitter identifies the coordinates provided by the user as occurring in a known place. Second, if the user manually selects the place when authoring the tweet.

Since coordinates give a more precise location, we use them instead of place when available. If we only have a place, we assume that the user is in the center of the place, as given by the place's bounding box.

For points of interest and neighborhoods, Twitter only provides the country in the associated metadata. While in some cases the city can be

---

[1]https://developer.twitter.com/en/docs/tweets/filter-realtime/overview/statuses-filter

parsed from the name, and the state inferred, we opted to exclude these places from our analysis for states. The full location details can be obtained from querying the Twitter API, but the magnitude of data in our analysis made this too time consuming. This excluded about 1.8% of our data.

We include an analysis of the 50 most populous United States cites. For this analysis, we included points of interest that had the city name in their names, e.g. "New York City Center". Specifically for New York City, we include places that corresponded to each of the five New York City boroughs (Brooklyn, Manhattan, Queens, Staten Island, The Bronx).

In summary, for each geolocated tweet we have an associated latitude and longitude.

## 4 Computing Mobility

We define the Twitter Social Mobility Index as follows. For each user, we collect all locations (coordinates) in a one week period, where a week starts on Monday and ends the following Sunday. We compute the centroid of all of the the coordinates and consider this the "home" location for the user for that week. We then measure the distance between each location and the centroid for that week. For distance, we measure the geodesic distance in kilometers between two adjacent records using geopy[2]. After collecting the distances we measure the standard deviation of these distances. In summary, this measure reflects the area and regularity of travel for a user, rather than the raw distance traveled. Therefore, a user who takes a long trip with a small number of checkins would have a larger social mobility measure than a user with many checkins who traveled in a small area. As the measure is sensitive to the number of checkins, it would reflect when people has less checkins during the pandemic.

We aggregate the results by week by taking the mean measure of all users in a given geographic area. We also present results for a 7-day moving average aggregation as a measure of daily movement. We record the variance of these measures to study the travel variance in the population, which will indicate if travel is reduced overall but not for some users.

We produce aggregate scores by geographic area for the United State as a whole, for each US state and territory, and for the 50 most populous cities in the US. We determine the geographic area of a

---

[2]https://github.com/geopy/geopy

user based on their centroid location for all time in our collection.

We compute the social mobility index for each day and week between January 1, 2019 and April 27, 2020. We select the date of March 16, 2020 as the start of social distancing on the national level, though individual states have implemented practices at different times. Therefore, we divide the data into two time periods: before social distancing (January 1, 2019 - March 15, 2020) and after social distancing (March 16th, 2020 - April 27, 2020).

We then compute the group level reduction in social mobility by considering average values as follows:

$$\text{Mobility Reduction} = 1 - \frac{\text{mobility after social distancing}}{\text{mobility before social distancing}}. \tag{1}$$

We also compute the reduction for each user and then track the median value, number of users active in both periods, and proportion of active users that completely reduce their mobility. We also conduct a similar analysis for seasonal effects by comparing mobility after social distancing and mobility during same period in 2019.

To handle sparse data issues in our dataset, we exclude (1) users with less than 3 geolocated tweets overall, and (2) a weekly record for a user if that user has less than 2 geolocated tweets in that week. Additionally, due to data loss in our data collection process we remove two weeks with far less data than other time periods by taking a 99.75% confidence limit on number of users and records.

## 5 Results

**Social Mobility Index** Table 2 shows the Twitter Social Mobility Index measured in kilometers for every state and territory in United States, and United States as a whole. City results appear in Table 3. We also include the rank of location by the group level reduction.

A few observations. The overall drop in mobility across the United States was large: 61.83%. Figure 1 shows the weekly social mobility index for the United States for the entire time period of our dataset. The figure reflects a massive drop in mobility starting in March, with the four most recent weeks the lowest on record in our dataset. Second, every US state and territory saw a drop in mobility, ranging from 38.54% to 76.80% travel compared to numbers before March 16, 2020. However, the

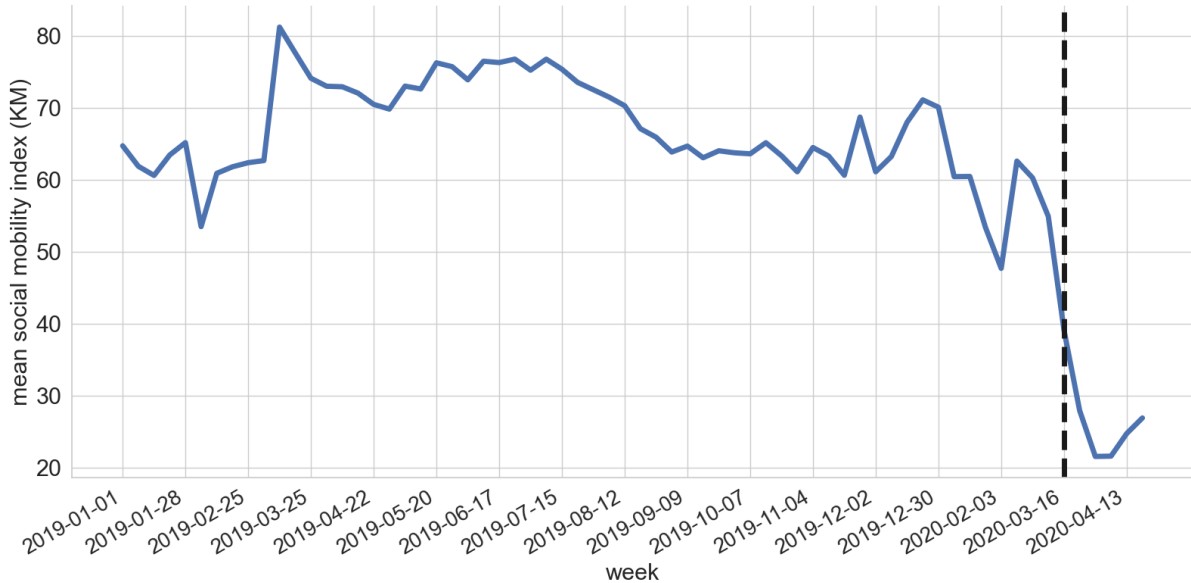

Figure 1: Mean social mobility index (KM) in United States from January 1, 2019 to April 27, 2020. Weeks with missing data are excluded from the figure.

variance by state was high. States that were early adopters of social distancing practices are ranked highly on the reduction in travel: e.g. Washington (3) and Maryland (9). In contrast, the eight states that do not have state wide orders as of the start of April (Zeleny, 2020) rank poorly: Arkansas (45), Iowa (37), Nebraska (35), North Dakota (22), South Carolina (38), South Dakota (46), Oklahoma (50), Utah (14), Wyoming (53). We observe similar trends in the city analysis, but the median users in these cities have a larger mobility reduction than the ones in the states.

Besides the group level mobility reduction (Eq. 4), we also examine the distribution of user level reduction. We only consider users that have at least two checkins in both periods, leading to a subgroup of all the users in the dataset for the reduction distribution. The median values for the reduction distribution is close to $100\%$ for most states. The median values for seasonal reduction are all smaller, but still suggest that people substantially reduce their mobility during the pandemic. Moreover, in the United States, $40\%$ of the 818,213 active users completely reduced their mobility, i.e., mobility reduction of $100\%$. In contrast, the same period in 2019 saw a $31\%$ reduction among 286,217 active users.

The White House announced "Slow the Spread" guidelines for persons to take action to reduce the spread of COVID-19 on March 16, 2020. $49.06\%$ of the states had their largest mobility drop in the week March 16 - 22, 2020 and $22.64\%$ in the following week. We compute a moving-average of daily mobility data, and use an offline change point detection method (Truong et al., 2020) on this trend. $62.26\%$ of the change points in 2020 are after the national announcement date but before the dates when individual state policies were enacted. This suggest that the national announcement had the largest effect as compared to state policies, a similar finding to the cell-phone-based mobility analysis of four large cities (Lasry et al., 2020). We also observe that, among 40 states that have announced Stay at Home policy, $92.5\%$ of the states have a more stationary daily mobility time series before the policy-announced date, compared to the mobility time series over all time, suggesting a rapid mobility change during pandemic.

Finally, Figure 2 shows a box-plot of the mobility variance across all users in a given time period. The distribution is long-tailed with a lot zeros, so we take the log of 1 plus each mobility index. While mobility is reduced in general, some users are still showing a lot of movement, suggesting that social distancing is not being uniformly practiced. These results clearly demonstrate that our metric can track drops in travel, suggesting that it can be used as part of ongoing pandemic response planning.

**Correlation** What are some of the factors that may help explain our Twitter Social Mobility In-

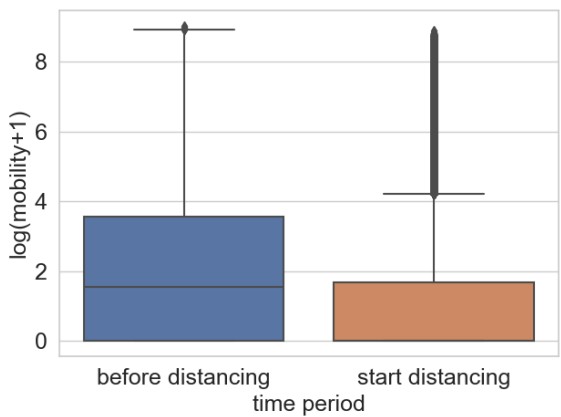

Figure 2: User distribution of mean social mobility index (KM) before/after social distancing in United States.

| Policy | Correlation |
|---|---|
| State of emergency | 0.2587 |
| Date banned visitors to nursing homes | 0.1510 |
| Stay at home/ shelter in place | 0.1507 |
| Froze evictions | 0.1411 |
| Closed non-essential businesses | 0.1359 |
| Closed gyms | 0.0765 |
| Closed movie theaters | 0.0737 |
| Closed day cares | 0.0563 |
| Closed restaurants except take out | 0.0341 |
| Date closed K-12 schools | -0.0821 |

Table 1: Pearson correlation between cumulative confirmed COVID-19 cases at May 10, 2020 and policy release date at each state.

dex? How well does the index track COVID-19 cases compared to other relevant factors? We analyze our data using a correlation analysis. We compute daily infection rate by dividing the number of new confirmed COVID-19 cases in each US state[3] by the population of the state. We compare the daily infection rate with social mobility index and the following trends (Raifman et al., 2020).

- The size of the state in square miles.

- The number of homeless individuals (2019).

- The unemployment rate (2018)

- The percentage of the population at risk for serious illness due to COVID-19.

For each day we compute the correlation between the daily infection rate and the above data by state.

Figure 3 shows the correlation by day. We adopt infection rate because raw confirmed cases is not as informative as the population has the highest correlation. However, the most significant factor in the early stage are still population related factors, i.e., number of homeless. We don't see significant correlations with other factors including the social mobility index. Starting from mid-March, we observe trends that unemployment rate, size of the state and social mobility index have increasing correlation but still not significant enough (the absolute correlation values < 0.5). The fluctuation in the middle is when states started to report confirmed cases.

We conduct a similar correlation analysis between each data source and the social mobility index, shown in Figure 4. As expected, Geographical state size has the highest positive correlation. We also observe that the number of people at risk for serious illness due to COVID-19 has negative correlation at the early stage of the pandemic.

Table 1 investigates the effect of various restriction policies on confirmed cases by running a similar correlation analysis on cumulative confirmed cases for each state on May 10, 2020. The policy types follow the data from (Raifman et al., 2020). We use the time difference (in days) between May 10, 2020 and policy-released date as the input for the analysis, and assign a negative value (-1000) for states that haven't announced the policy. The factor with the highest correlation to the social mobility index is the declaration of a state of emergency, which is the broadest type of policy.

## 6 Related Work

There is a long line of work on geolocation prediction for Twitter, which requires inferring a location for a specific tweet or user (Dredze et al., 2013; Zheng et al., 2018; Han et al., 2014; Pavalanathan and Eisenstein, 2015). This includes work on patterns and trends in Twitter geotagged data (Dredze et al., 2016c). While most of this work focused on a user, and thus is not suitable for tracking a user's movements, there may be opportunities to combine these methods with our approach.

There have been many studies that have analyzed Twitter geolocation data to study population movements. Hawelka et al. (2014) demonstrated a method for computing global travel patterns from Twitter, and Dredze et al. (2016b) adapted this

---

[3] https://github.com/CSSEGISandData/COVID-19

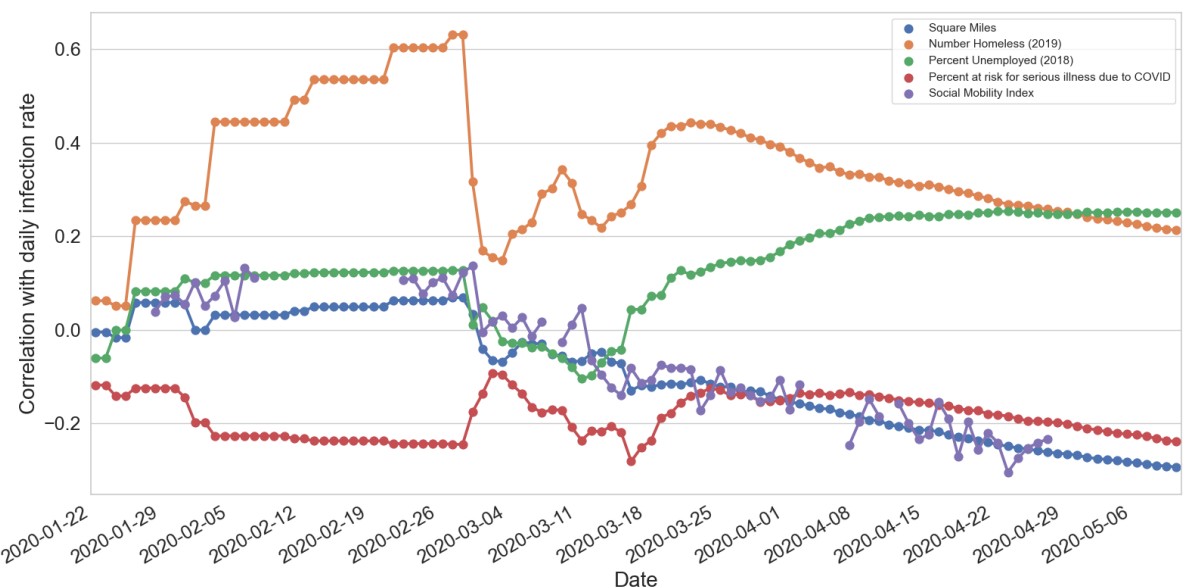

Figure 3: Pearson correlation between daily COVID-19 infection rates and various factors at state level.

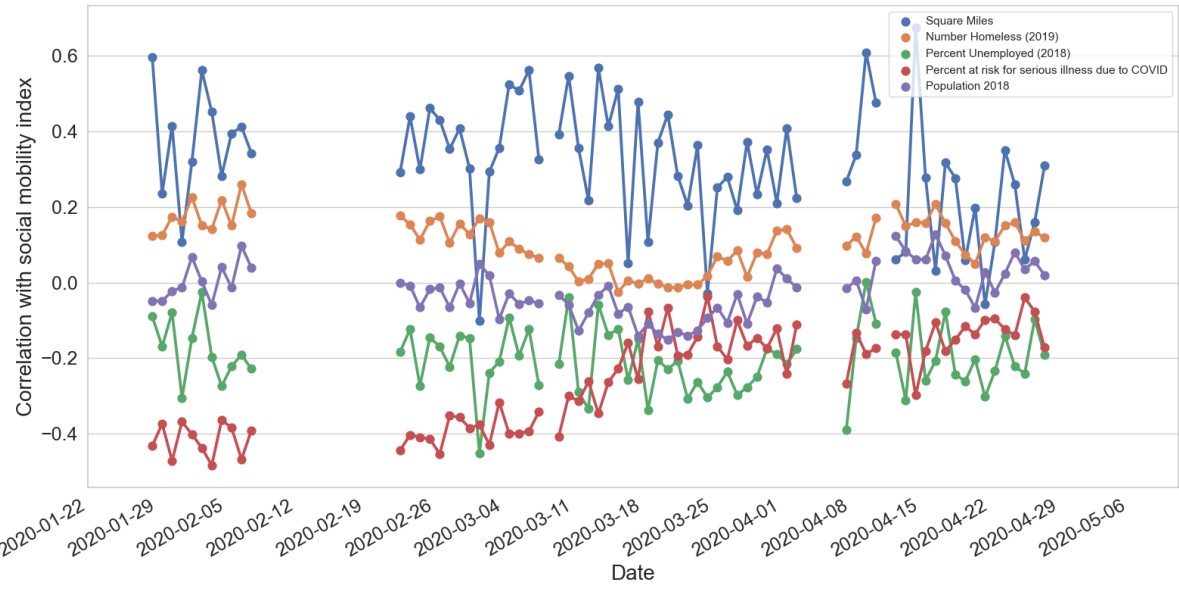

Figure 4: Pearson correlation between social mobility index and various factors at state level.

method to support efforts in combating the Zika epidemic.

Several studies have used human mobility patterns from Twitter data (Jurdak et al., 2015; Huang and Wong, 2015; Birkin et al., 2014; Hasan et al., 2013). These studies have included analyses of urban mobility patterns (Luo et al., 2016; Soliman et al., 2017; Kurkcu et al., 2016). Finally, some of these analyses have considered mobility patterns around mass events (Steiger et al., 2015).

## 7 Conclusion

We presented the Twitter Social Mobility Index, a measure of social mobility based on public Twitter geolocated tweets. Our analysis shows that overall in the United States there has been a large drop in mobility. However, the drop is inconsistent and varies significantly by state. It appears that states that were early adopters of social distancing practices have more significant drops than states that have not yet implemented these practices.

Our work on this data is ongoing, and there are several directions that warrant further study. First,

as states begin to reopen, and some states maintain restrictions, tracking changes in population behaviors will be helpful in making policy decisions. Second, we focused on the United States, but Twitter data provides sufficient coverage for many countries to replicate our analysis. Third, for each user in the dataset there exists tweet content, that can reflect a user's attitudes, beliefs and behaviors. Studying these together with their mobility reduction could yield further insights. Our findings are presented on `http://socialmobility.covid19dataresources.org` and we will continue to update our analysis during the pandemic.

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

| location | Mobility (KM) | | | User level reduction | | Rank |
|---|---|---|---|---|---|---|
| | Before distancing | After distancing | Group level reduction | Median reduction | Median seasonal reduction | |
| AK | 109.76 | 25.47 | 76.80% | 99.84% | 63.73% | 1 |
| AL | 48.04 | 22.57 | 53.03% | 84.47% | 72.94% | 47 |
| AR | 50.54 | 23.15 | 54.19% | 91.87% | 76.81% | 45 |
| AZ | 62.85 | 23.47 | 62.66% | 93.69% | 85.55% | 26 |
| CA | 78.58 | 29.60 | 62.33% | 96.65% | 91.35% | 29 |
| CO | 72.23 | 24.47 | 66.12% | 98.23% | 93.37% | 12 |
| CT | 45.51 | 14.89 | 67.28% | 96.29% | 89.25% | 8 |
| DC | 77.67 | 19.74 | 74.58% | 100.00% | 97.75% | 2 |
| DE | 43.63 | 13.61 | 68.81% | 93.44% | 85.08% | 7 |
| FL | 76.99 | 32.24 | 58.13% | 92.38% | 82.92% | 42 |
| GA | 65.64 | 27.11 | 58.70% | 85.26% | 78.00% | 39 |
| HI | 147.61 | 70.75 | 52.07% | 97.69% | 89.21% | 51 |
| IA | 50.42 | 20.59 | 59.17% | 95.91% | 89.82% | 37 |
| ID | 70.77 | 33.36 | 52.86% | 94.12% | 78.19% | 49 |
| IL | 55.59 | 19.38 | 65.15% | 98.71% | 93.01% | 16 |
| IN | 45.86 | 17.15 | 62.60% | 97.19% | 89.61% | 27 |
| KS | 65.50 | 23.19 | 64.60% | 97.03% | 81.57% | 19 |
| KY | 44.67 | 15.31 | 65.74% | 93.93% | 83.42% | 13 |
| LA | 45.98 | 19.39 | 57.83% | 86.13% | 77.76% | 43 |
| MA | 58.69 | 17.64 | 69.95% | 98.83% | 93.93% | 5 |
| MD | 46.10 | 15.19 | 67.04% | 94.80% | 88.67% | 9 |
| ME | 59.68 | 22.45 | 62.38% | 93.77% | 78.53% | 28 |
| MI | 56.24 | 20.96 | 62.72% | 96.84% | 90.42% | 25 |
| MN | 64.01 | 21.68 | 66.13% | 98.36% | 91.34% | 11 |
| MO | 52.27 | 20.08 | 61.59% | 95.89% | 88.65% | 31 |
| MS | 50.24 | 24.36 | 51.51% | 79.09% | 69.11% | 52 |
| MT | 69.93 | 32.96 | 52.86% | 90.17% | 65.58% | 48 |
| NC | 52.11 | 19.73 | 62.14% | 94.27% | 85.26% | 30 |
| ND | 65.77 | 23.65 | 64.04% | 99.71% | 97.21% | 22 |
| NE | 55.11 | 21.88 | 60.29% | 99.95% | 91.40% | 35 |
| NH | 55.09 | 19.48 | 64.64% | 96.26% | 85.35% | 18 |
| NJ | 49.27 | 14.62 | 70.33% | 97.28% | 93.41% | 4 |
| NM | 58.20 | 24.23 | 58.37% | 95.66% | 73.14% | 41 |
| NV | 80.25 | 33.19 | 58.64% | 93.42% | 85.00% | 40 |
| NY | 71.17 | 24.57 | 65.48% | 98.94% | 94.20% | 15 |
| OH | 44.88 | 15.73 | 64.95% | 94.81% | 88.68% | 17 |
| OK | 52.34 | 24.69 | 52.83% | 88.38% | 76.99% | 50 |
| OR | 71.12 | 25.97 | 63.49% | 97.51% | 92.68% | 24 |
| PA | 54.40 | 19.45 | 64.24% | 97.59% | 89.85% | 20 |
| PR | 44.96 | 14.94 | 66.77% | 97.26% | 90.38% | 10 |
| RI | 46.80 | 14.50 | 69.01% | 96.74% | 90.55% | 6 |
| SC | 48.28 | 19.85 | 58.88% | 86.03% | 77.92% | 38 |
| SD | 68.41 | 31.52 | 53.92% | 95.91% | 86.66% | 46 |
| TN | 56.77 | 21.83 | 61.55% | 94.89% | 85.89% | 32 |
| TX | 73.24 | 28.60 | 60.95% | 93.81% | 84.18% | 34 |
| UT | 68.43 | 23.62 | 65.49% | 93.56% | 91.50% | 14 |
| VA | 57.37 | 22.33 | 61.07% | 95.62% | 87.51% | 33 |
| VI | 132.16 | 47.57 | 64.00% | 98.66% | 87.72% | 23 |
| VT | 56.84 | 20.33 | 64.23% | 96.35% | 86.70% | 21 |
| WA | 75.34 | 21.31 | 71.71% | 98.43% | 95.72% | 3 |
| WI | 56.32 | 22.68 | 59.74% | 96.88% | 91.75% | 36 |
| WV | 46.59 | 20.02 | 57.02% | 88.95% | 82.40% | 44 |
| WY | 71.64 | 44.03 | 38.54% | 84.95% | 50.90% | 53 |
| United States | 65.59 | 25.04 | 61.83% | 95.86% | 88.36% | - |

Table 2: Reduction of mobility for all states and territories in United States and United States. Ranks are based on group level reduction.

| | Mobility (KM) | | | User level reduction | | |
|---|---|---|---|---|---|---|
| location | Before distancing | After distancing | Group level reduction | Median reduction | Median seasonal reduction | Rank |
| New York City | 86.37 | 29.91 | 65.38% | 99.70% | 96.69% | 27 |
| Los Angeles | 103.16 | 40.86 | 60.39% | 98.69% | 93.87% | 40 |
| Chicago | 64.09 | 19.87 | 69.00% | 99.96% | 94.58% | 14 |
| Houston | 53.70 | 21.50 | 59.96% | 97.04% | 88.00% | 41 |
| Phoenix | 60.07 | 19.12 | 68.17% | 96.32% | 91.08% | 18 |
| Philadelphia | 54.80 | 17.70 | 67.71% | 99.16% | 93.70% | 19 |
| San Antonio | 45.43 | 15.93 | 64.93% | 99.00% | 91.33% | 28 |
| San Diego | 79.21 | 28.19 | 64.41% | 98.67% | 92.77% | 30 |
| Dallas | 63.92 | 21.85 | 65.81% | 95.48% | 89.32% | 25 |
| San Jose | 60.63 | 14.82 | 75.55% | 99.88% | 97.34% | 2 |
| Austin | 72.50 | 22.84 | 68.50% | 99.66% | 94.66% | 17 |
| Jacksonville | 47.06 | 26.87 | 42.90% | 96.60% | 92.92% | 50 |
| Fort Worth | 51.67 | 19.68 | 61.92% | 95.33% | 85.72% | 37 |
| Columbus | 44.67 | 14.73 | 67.02% | 96.91% | 93.15% | 22 |
| San Francisco | 113.77 | 31.99 | 71.89% | 99.93% | 98.94% | 8 |
| Charlotte | 58.13 | 20.90 | 64.04% | 96.26% | 89.83% | 31 |
| Indianapolis | 46.50 | 14.53 | 68.76% | 99.26% | 91.85% | 15 |
| Seattle | 98.92 | 21.64 | 78.12% | 99.98% | 99.06% | 1 |
| Denver | 81.11 | 23.08 | 71.55% | 99.05% | 96.30% | 9 |
| Washington | 80.26 | 22.12 | 72.43% | 99.93% | 97.27% | 7 |
| Boston | 77.58 | 27.47 | 64.59% | 99.42% | 96.40% | 29 |
| El Paso | 51.10 | 21.50 | 57.92% | 100.00% | 95.97% | 44 |
| Detroit | 53.94 | 22.38 | 58.50% | 94.89% | 83.68% | 43 |
| Nashville | 72.83 | 23.94 | 67.13% | 98.45% | 94.88% | 21 |
| Portland | 78.91 | 24.81 | 68.56% | 99.45% | 96.81% | 16 |
| Memphis | 48.64 | 18.41 | 62.15% | 98.65% | 86.75% | 35 |
| Oklahoma City | 46.07 | 16.78 | 63.57% | 91.34% | 75.19% | 33 |
| Las Vegas | 80.21 | 35.69 | 55.50% | 94.87% | 83.90% | 47 |
| Louisville | 45.52 | 12.97 | 71.51% | 94.31% | 77.68% | 10 |
| Baltimore | 45.61 | 11.66 | 74.43% | 96.10% | 89.37% | 4 |
| Milwaukee | 52.01 | 22.78 | 56.19% | 97.01% | 91.86% | 46 |
| Albuquerque | 51.04 | 16.88 | 66.93% | 98.95% | 75.81% | 23 |
| Tucson | 53.58 | 23.10 | 56.89% | 95.73% | 84.48% | 45 |
| Fresno | 37.39 | 10.84 | 71.02% | 96.06% | 89.20% | 11 |
| Mesa | 48.77 | 21.72 | 55.47% | 92.40% | 71.33% | 48 |
| Sacramento | 62.14 | 25.45 | 59.05% | 94.82% | 94.47% | 42 |
| Atlanta | 87.90 | 33.39 | 62.02% | 93.50% | 86.36% | 36 |
| Kansas City | 62.93 | 17.23 | 72.61% | 98.30% | 96.54% | 6 |
| Colorado Springs | 64.82 | 23.55 | 63.67% | 99.47% | 95.66% | 32 |
| Miami | 114.33 | 55.77 | 51.22% | 97.55% | 88.56% | 49 |
| Raleigh | 51.62 | 15.24 | 70.47% | 97.79% | 89.51% | 12 |
| Omaha | 49.99 | 15.38 | 69.24% | 100.00% | 93.72% | 13 |
| Long Beach | 54.97 | 20.51 | 62.70% | 93.33% | 89.75% | 34 |
| Virginia Beach | 48.91 | 18.92 | 61.33% | 96.35% | 88.38% | 39 |
| Oakland | 87.36 | 22.26 | 74.52% | 98.41% | 96.26% | 3 |
| Minneapolis | 69.67 | 18.72 | 73.14% | 99.14% | 94.21% | 5 |
| Tulsa | 48.54 | 18.51 | 61.85% | 99.89% | 93.20% | 38 |
| Arlington | 56.42 | 18.27 | 67.62% | 97.58% | 93.25% | 20 |
| Tampa | 70.50 | 23.55 | 66.59% | 94.48% | 83.23% | 24 |
| New Orleans | 55.96 | 19.18 | 65.73% | 97.00% | 88.75% | 26 |

Table 3: Reduction of mobility for top 50 United States cities by population. Ranks are based on group level reduction.