# OpenReview forum: "The Twitter Social Mobility Index: Measuring Social Distancing Practices from Geolocated Tweets"
_aclweb.org/ACL/2020/Workshop/NLP-COVID — Submitted to NLP-COVID-2020_

### Official Review · AnonReviewer3 · 2020-06-08
**Seems like a solid study, but out of scope (no NLP)**

**Rating:** 3
**Confidence:** 4

**Review:**

Summary: geo-location data from tweets is used to measure mobility, which is compared over time and over locations.

The manuscript is well written and makes for a clear case. It would be a good submission for some other workshop or conference, but is out of scope for this particular workshop. Even with the broad scope defined in the call for submissions ("any aspect of natural language processing (NLP) applied to combat the COVID-19 pandemic"), this falls outside as there is no aspect of natural language processing. Just as one indication: neither of the terms 'language' or 'NLP' occurs in the body of the submission.

The rest of the material appears interesting and well executed, but validity of the approach was not assessed by this reviewer.

---

### Decision · Program_Chairs · 2020-06-10

**Decision:**

Reject

**Comment:**

Unfortunately as this work does not involve analysis of the content of the Twitter tweets, this is out of scope for the workshop.

Best of luck to you.